# Current Status and Influencing Factors of Eating Behavior in Residents at the Age of 18~60: A Cross-Sectional Study in China

**DOI:** 10.3390/nu14132585

**Published:** 2022-06-22

**Authors:** Dongli Mei, Yuqian Deng, Qiyu Li, Zhi Lin, Huiwen Jiang, Jingbo Zhang, Waikit Ming, Hao Zhang, Xinying Sun, Guanyun Yan, Yibo Wu

**Affiliations:** 1School of Nursing, Peking University, Beijing 100191, China; 2011210175@stu.pku.edu.cn; 2Xiangya School of Nursing, Central South University, Changsha 410000, China; dyq2468518354@163.com; 3School of Humanities and Health Management, Jinzhou Medical University, Jinzhou 121000, China; liqiyu20200907@163.com; 4College of Communication and Art Design, University of Shanghai for Science and Technology, Shanghai 200000, China; z13713965329@163.com; 5Department of Public Administration, School of Health Administration, Harbin Medical University, Harbin 150086, China; j2770402762@163.com; 6School of Humanities and Social Sciences, Harbin Medical University, Harbin 150081, China; zjb952528@163.com; 7Department of Infectious Diseases and Public Health, Jockey Club College of Veterinary Medicine and Life Sciences, City University of Hong Kong, Hong Kong 999077, China; wkming2@cityu.edu.hk; 8School of Pharmacy, Bengbu Medical University, Bengbu 233000, China; 13955266068@163.com; 9Department of Pharmacy, First Affiliated Hospital of Bengbu Medical University, Bengbu 233000, China; 10School of Public Health, Peking University, Beijing 100191, China; xysun@bjmu.edu.cn

**Keywords:** eating behavior, influencing factor, overweight, obesity, Ecological Model of Health Behavior, a cross-sectional study

## Abstract

As eating behavior is important to health, this cross-sectional study was conducted to analyze the factors influencing the eating behavior related to overweight and obesity of Chinese residents aged 18~60 based on the Ecological Model of Health Behavior. The short-form of the Eating Behavior Scale (EBS-SF) was applied to evaluate eating behavior. The multivariable linear stepwise regression analysis was used to identify and analyze the influence factors, and the receiver operating characteristic curves analysis to validate the predictive capability of the EBS-SF score in differentiating overweight and obesity. A total of 8623 participants were enrolled. In the personal characteristics, male (β = −0.03), older [36–45 years (β = −0.06) or 46–60 years (β = −0.07)], higher scores of Agreeableness (β = −0.04), Conscientiousness (β = −0.14) or Openness (β = −0.03) contributed to healthy eating behavior. In the individual behaviors, those who smoked (β = 0.04), drank alcohol (β = 0.05), exercised frequently (β = 0.07), had higher PHQ-9 scores (β = 0.29) may have improper eating habits. As for the interpersonal networks, the residents who were married (β = −0.04) behaved well when eating, while those who had offspring or siblings tended to have unhealthy eating behavior. At the community level, living in Western China (β = −0.03), having a monthly household income of 6001–9000 yuan per capita (β = −0.04), having no debt (β = −0.02), being retired (β = −0.03), or having lower PSSS scores (β = −0.03) led to lower EBS-SF scores. And the EBS-SF score demonstrated a moderate-high accuracy in predicting overweight and obesity.

## 1. Introduction

Eating behavior is closely related to body weight and thus has a great impact on health. According to the latest data from the “Report on the Nutrition and Chronic Diseases of Chinese Residents (2020)”, more than half of Chinese adults were overweight or obese. And the overweight rate and obesity rate of adult residents (≥18 years old) reached 34.3% and 16.4% respectively, which is mainly due to uncontrolled eating behavior [1]. There is a large amount of data showing that unhealthy eating behaviors, such as emotional eating and binge eating, are significant factors leading to excess weight, while healthy eating behaviors, including having balanced diets, can reduce the risk of diet-related chronic diseases, such as cardiovascular disease, diabetes, and some cancers [2,3,4]. In addition, healthy eating behavior plays an important role in the well-being and the financial impact of the whole society [5]. Therefore, it is essential to explore the related influencing factors of eating behavior and then identify the intervention in eating behavior. Further investigations are required to promote healthy eating behavior, possibly decreasing the incidence of overweight and obesity among residents.

Not only is eating to obtain adequate nutrients to satisfy biological requirements, but eating behavior is also dictated by a complex interaction between individuals, interpersonal networks, and social environment [6]. Sex, age, and physical condition are associated with food cravings and then influence eating behavior [7]. Personality traits have an impact on eating styles and food choices, leading to differences in eating behavior among different people [8]. Self-efficacy, referring to the set of beliefs residents hold about their ability to complete a particular task, also has a significant effect on eating behavior [9]. Moreover, individual behaviors, including smoking, drinking, exercise, and emotional regulation, influence eating behavior by changing several factors, such as caloric intake, metabolic rate, and lipogenesis [10,11,12,13]. Interpersonal relationships serve as a significant factor influencing eating behavior through social influence, and bad interpersonal relationships are even one of the main factors for eating disorders [14,15]. There exists an important influence on eating behavior among family members such as spouses, offspring, and siblings [16,17,18]. Additionally, there is ample evidence that social and community environment, including residence, education, career, and income, has a powerful effect on eating behavior such as food choice and amounts consumed [19,20]. However, the influence factors of eating behavior related to being overweight and obese and the effect of the influence factors on the eating behavior remains not fully understood [6].

## 2. Theoretical Basis

The Ecological Model is widely used to analyze questions about health, especially in health intervention and health promotion. Considering multiple dimensions of influence, the Ecological Model of Health Behavior pays direct attention to both health behavior and its individual and environmental determinants. It emphasizes that the health of individuals or groups is the result of the joint action of themselves and the living environment, helping develop more comprehensive interventions for health behavior [21,22]. There are five levels of the Ecological Model of Health Behavior, including personal characteristics, individual behaviors, interpersonal networks, community, and public policy, of which the level of the personal characteristics includes factors such as sex, age, personality traits, and self-efficacy. The level of individual behaviors focuses on lifestyle and emotion-processing, while the level of interpersonal networks concerns social support, and marital and family status. The community level includes living conditions, career status, and social-economic status, and the public policy level contains all regulatory legislature. The main factors in each level of the Ecological Model of Health Behavior are shown in Figure 1 (see Figure 1). Each of the levels is an open system, interacting with and interdependent on each other in the model. The Ecological Model was widely adopted to analyze the influencing factors of specific health behavior. Sogari et al. used the Ecological Model to analyze the eating habits of American college students [23]. Thurston et al. argued that the Ecological Model that incorporated the social, psychological, and biological could explain the influencing factors of immigrant women’s health [24]. In this study, the first four levels, including personal characteristics, individual behaviors, interpersonal networks, and community, were considered when using the Ecological Model of Health Behavior to analyze the influencing factors.

## 3. Research Purpose and Hypotheses

At present, some foreign scholars have performed research on the influence of genetics, physiology, lifestyle, and environmental factors related to food characteristics on eating behavior [25,26,27]. Domestically, a series of studies were carried out on eating behavior and factors affecting specific groups of people, such as middle school students and young children [28,29]. However, previous studies mostly focused on general eating behavior and few specifically explored the influencing factors of eating behavior related to overweight and obesity. Moreover, the effect of factors on eating behavior was debated [6]. Therefore, this study was conducted to identify the factors affecting the eating behavior related to overweight and obesity of residents aged 18~60 in China and analyzed the effects of the influence factors on eating behavior according to the Ecological Model of Health Behavior, with a view to providing reference and feasible suggestions for reducing the incidence of overweight and obesity and promoting healthy eating behavior of residents.

Accordingly, we put forward the following assumptions in the present study. As shown in Figure 1, H1–H14 are fourteen hypotheses of the relationship between influencing factors and eating behavior related to overweight and obesity in this study (see Figure 1):

I: In the first level, sex (H1), age (H2), and personality traits (H3) have complex effects on eating behavior, while self-efficacy (H4) has a positive impact on eating behavior related to overweight and obesity.

II: In the second level, smoking (H5) and drinking alcohol (H6) are negatively associated with healthy eating behavior, exercise (H7) has a complex impact on eating behavior due to the complex physiological mechanisms of metabolic rate, whereas emotional regulation (H8) contributes to better eating behavior.

III: In the third level, marital status (H9) and family environment (H10) have complex impacts on eating behavior, while social support (H11) has a positive effect on better eating behavior.

IV: In the fourth level, a good community environment, including better living conditions (H12), stabler career status (H13), and higher social-economic status (H14), is beneficial to eating behavior.

## 4. Materials and Methods

### 4.1. Study Design

The study was conducted by multistage sampling from 10 July 2021 to 15 September 2021. Based on the Chinese population pyramid, quota sampling of the selected residents in 120 cities was conducted with the quota attributes of sex, age, and urban-rural distribution to obtain the samples by sex, age, and urban-rural distribution in line with the demographic characteristics [30]. The investigators or survey teams (≤10 people) were recruited openly and trained in the sample cities. At least one investigator or one investigation team was recruited in each city, each investigator was responsible for collecting 30–90 questionnaires, and each team was responsible for collecting 100–200 questionnaires.

The survey was carried out through the network Wenjuanxing platform, the most popular survey software in China (https://www.wjx.cn/ (accessed on 1 July 2021)), by investigators issuing questionnaires to residents one-on-one. The participants signed the informed consent form and answered the questionnaires by clicking on the link, and the investigators input the questionnaire number. If the respondent had the ability to think but did not have enough action ability to answer the questionnaire, the investigator would conduct a one-to-one interview and then answer the questions on his or her behalf.

### 4.2. Participants

Inclusion criteria: (1) Aged 18~60; (2) Had the nationality of the People’s Republic of China; (3) China’s permanent resident population with an annual travel time ≤ 1 month; (4) Participated in the study and fill in the informed consent form voluntarily; (5) Participants can complete the questionnaire survey by themselves or with the help of investigators; (6) Participants can understand the meaning of each item in the questionnaire.

The exclusion criteria were as follows: (1) Persons with unconsciousness, or mental disorders; (2) Those who were participating in other similar research projects.

Initially, 11,031 participants from 120 cities in the 23 provinces, 5 autonomous regions, and 4 municipalities finished the questionnaire. We included the residents at the age of 18~60, as the Eating Behavior Scale and some other items in the questionnaire were inapplicable to residents less than 18 or more than 60 years old. After excluding the results with missing data or logic errors, 8623 residents were enrolled in this study. Figure 2 shows a detailed flowchart of the enrollment (see Figure 2).

### 4.3. Instruments

The questionnaire consisted of two parts, focusing on the current status of residents’ eating behaviors and related influence factors. The first part was a self-made part that surveyed the social-demographic characteristics of residents such as sex, age, education level, career status, and marital status, as well as their basic family information such as family structure and family finances. The second part was a series of standard scales, including the short-form of the Eating Behavior Scale (EBS-SF), the 10-item short version of the Big Five Inventory (BFI-10), the New General Self-Efficacy Scale (NGSES), the Sport Scale, the Patient Health Questionnaire (PHQ-9) and the 7-item Generalized Anxiety Disorder (GAD-7), the short-form of the Family Health Scale (FHS-SF) and the Perceived Social Support Scale (PSSS).

#### 4.3.1. The Short-Form of the Eating Behavior Scale (EBS-SF)

The short-form of the Eating Behavior Scale (EBS-SF), designed to assess dietary behavior abnormalities related to obesity, is a reliable and valid measure as an indicator of obesity in both clinical and research settings [31]. The EBS-SF was originated from the 30-item Sakata Eating Behavior Scale (EBS), a widely validated and used scale, of which seven items are adopted as the short version [32]. The correlation between EBS-SF and the original EBS is extremely high (r = 0.93, *p* = 0.001) [32]. Considering the time and mental costs of the respondents, the short-form of the Eating Behavior Scale was applied in this study to assess the eating behavior of residents. The EBS-SF was scored on a 4-point Likert scale indicating the residents’ degree of agreement (1 = strongly disagree, 2 = somewhat disagree, 3 = somewhat agree, and 4 = strongly agree), and seven items were summed to obtain scores between 7 and 28, with higher scores indicating worse eating behavior. The Cronbach’s α of the EBS-SF in the present study was 0.871, showing that it had good internal consistency.

#### 4.3.2. The 10-Item Short Version of the Big Five Inventory (BFI-10)

For the past 30 years, the Big Five Inventory (BFI) with 44 short-phrase items as the most widely accepted contemporary model of personality was verified in several studies and its five-factor structure has been substantially replicated in several countries [33,34,35]. The 10-item short version of the Big Five Inventory (BFI-10) was an abbreviated scale of BFI with significant levels of reliability and validity in different versions [36,37,38]. Therefore, the BFI-10 was applied to measure the personality characteristics of residents, including Extraversion, Agreeableness, Conscientiousness, Neuroticism, and Openness, on a 5-point Likert-type scale ranging from 1 (totally disagree) to 5 (totally agree) [38]. The scores of Extraversion were summed of the scores of item 1R and item 6, the scores of Agreeableness were combined with the scores of item 2 and 7R, the scores of Conscientiousness as 3R and 8, Neuroticism as 4R + 9, and Openness as 5R + 10 (R = item is reversed-scored). Reliability levels of the BFI-10 proved satisfactory using Cronbach’s α analysis: Extraversion (α = 0.723), Agreeableness (α = 0.759), Conscientiousness (α = 0.786), Neuroticism (α = 0.753) and Openness to experience (α = 0.714) [39].

#### 4.3.3. The New General Self-Efficacy Scale (NGSES)

The Weight Efficacy Lifestyle Scale, a measurement of self-efficacy for controlled eating, contained subscales of negative emotions and social environment, which overlapped with several influence factors and standard scales in this study [40]. Moreover, the internal consistency of the Weight Efficacy Lifestyle Scale remained to be proved, thus the Weight Efficacy Lifestyle Scale was not applied in the current study. The New General Self-Efficacy Scale (NGSES) is an instrument with widely-proven validity and reliability to measure residents’ belief in their overall competence to perform in a variety of situations [9]. Therefore, the NGSES was used in the present study to assess the capacity of participants to cope with life’s demands, including the food demand. It consisted of only eight items, saving time and mental costs for the participants. Respondents rated the degree of agreement of each item on a 5-point Likert scale (1 = strongly disagree, 2 = disagree, 3 = neither disagree nor agree, 4 = agree, 5 = strongly agree), resulting in a score ranging from 8 to 40. The NGSES demonstrated good internal consistency with the Cronbach’s α of 0.944 in this study.

#### 4.3.4. The Sport Scale

Currently, there was no standard scale specifically designed to assess the frequency of physical activity. The Sport Scale, which originated from the first dimension of the Self-Management Scale, was applied to assess the exercise frequency in one week, including six items of body-building, walking, swimming, bicycle riding, exercise with equipment, and other aerobic exercises [41]. Each item was scored on the 5-point Likert scale indicating exercise frequency (0 = not doing, 1 = less than 30 min per week, 2 = 30~59 min per week, 3 = 1~3 h per week, and 4 = more than 3 h per week). All 6 items were summed to obtain scores from 0 to 24, with higher scores indicating more exercise frequencies. The Cronbach’s α of 0.811 proved the satisfactory reliability level of the Sport Scale, showing the Sport Scale can be used to assess the exercise frequency of residents.

#### 4.3.5. The Patient Health Questionnaire-9 (PHQ-9)

The Patient Health Questionnaire-9 (PHQ-9) was designed to screen for depression complying with the criteria of the Diagnostic and Statistical Manual of Mental Disorders and was identified as the most reliable screening tool [42,43]. Each item of the nine-item questionnaire was scored on the frequency from “not at all” to “nearly every day”. The total score ranged from 0~27, with higher scores indicating more severe depression. A score between 0 and 4 indicated no symptoms, 5~9 indicated mild, 10~14 indicated moderate, 15~19 indicated moderately severe, and 19~27 indicated severe symptoms. The Cronbach’s α of the PHQ-9 was 0.939.

#### 4.3.6. The 7-Item Generalized Anxiety Disorder (GAD-7)

As a practical self-report anxiety questionnaire, the 7-item Generalized Anxiety Disorder (GAD-7) has been proven to be reliable and valid in the general population [44,45]. In this study, the GAD-7 was applied as a brief screening tool to detect anxiety, using seven items scored on a 4-point Likert scale indicating the frequency from “not at all” to “nearly every day” [45]. Ranging from 0 to 21, The higher the total scores, the more severe the symptoms of anxiety. A score of 0~4 indicated no symptoms, 5~9 mild symptoms, 10~14 moderate symptoms, and 15–21 severe symptoms. In addition, the Cronbach’s α was 0.954 in this study.

#### 4.3.7. The Short-Form of the Family Health Scale in the Chinese Version (FHS-SF)

The health of a family was influenced by multiple factors, such as family functioning, emotional support, economic resources, and so on. The short-form of the Family Health Scale (FHS-SF) was compiled by Crandall and Weiss to fully understand family health [46]. The FHS-SF was composed of 2~3 items with a large factor load and weight from the four dimensions of family social and emotional health process, family healthy lifestyle, family health resources, and family external social support in the long-form of the Family Health Scale (FHS), with a total of 10 items. A 5-point Likert scoring method was adopted for each item, and reverse scoring was adopted for questions 6, 9, and 10. The higher the score, the higher the family health level. The FHS-SF demonstrated good internal consistency with the Cronbach’s α of 0.850.

#### 4.3.8. The Perceived Social Support Scale (PSSS)

To measure social support from family, friends, relatives, and colleagues, the Perceived Social Support Scale (PSSS) has been widely used in a diverse population [47,48]. It was a 12-item instrument with each item rated on a 7-point Likert scale ranging from “strongly disagree” to “strongly agree” [49]. The items were summed to obtain scores between 12 and 84, with higher scores indicating higher perceived social support. In addition, the Cronbach’s α in this study was 0.959.

### 4.4. Quality Control

The short-form of the Eating Behavior Scale was used as an instrument of eating behavior associated with overweight and obesity in this study. However, the predictive power of the short-form of the Eating Behavior Scale in differentiating overweight and obesity remained unknown. Hence, we further analyzed and validated the predictive capability of the EBS-SF score in differentiating overweight and obesity in the Chinese population to ensure a reliable and convenient construct.

### 4.5. Statistical Methods

Data entry and analysis were performed using SPSS™ for Windows (version 27.0) (SPSS Inc., Chicago, IL, USA). The quantity and percentage of categorical variables, as well as the mean and standard deviation of continuous variables, were calculated using descriptive statistics. Student’s *t*-test and one-way ANOVA were used to compare differential factors for the EBS-SF score as appropriate. The multivariable linear stepwise regression analysis model was applied to estimate factors associated with the EBS-SF score in residents. As a step-by-step iterative construction of a regression model, the stepwise regression in this study involved removing potential explanatory variables in succession and testing for statistical significance after each iteration. The study sample was stratified by sex, region, and residence to conduct the regression analysis. In the regression analysis, the score of the EBS-SF was taken as the dependent variable. The independent variables mainly included the factors of each level of the Ecological Model of Health Behavior, among which the unordered multiple categorical variables and ordered multiple categorical variables were converted into multiple dummy variables. The dummy coding of categorical variables is listed in Table A1 in Appendix A. Combined with professional knowledge, the variance inflation factor (VIF) was used to detect multicollinearity. The VIF of the variables was all <5 with the mean VIF as 2.14, indicating that there was no obvious collinearity among the variables. The VIF of variables was presented in Table A2 in Appendix A. The R^2^ values were also calculated to determine how well the variances of the EBS-SF score were explained by the influencing factors. In addition, to validate the prediction accuracy of the EBS-SF score in differentiating overweight and obesity, the receiver operating characteristic (ROC) curves analysis was performed and the area under the curve (AUC) was calculated. A *p*-value of 0.05 or less was considered significant.

### 4.6. Ethics Statement

This study scheme has been approved by the Institutional Review Committee of Ji’nan University, Guangzhou, China (JNUKY-2021-018). All the participants fully understood the study and voluntarily signed informed consent forms.

## 5. Results

### 5.1. The Statistical Description of the EBS-SF Scores and Influencing Factors

The demographic information and statistical description of the participants were presented in Table 1 and Table 2. In total, 8623 participants were finally recruited into the study, among which 3894 people were male (45.2%) and 4729 were female (54.8%). There were 1996 participants (23.1%) aged 18 to 25 years old, and 2461 residents (28.5%) were aged from 26 to 35. 1998 respondents (23.2%) were 36–45 years old. The age of other 2168 people (25.1%) ranged from 46 to 60 years. More than half of the participants (51.4%) were from Eastern China, while 2217 (25.7%) and 1975 (22.9%) came from Central China and Western China, respectively. In addition, most respondents (74.8%) lived in urban areas, and the other residents (25.2%) lived in rural areas. The average score of EBS-SF was 16.66. The respondents who were male (16.72 ± 4.60), aged 18–35 [18–25 years (17.71 ± 4.55), 26–35 years (17.02 ± 4.54)], lived alone (17.68 ± 4.83), had a college degree or above [junior college (16.72 ± 4.55), bachelor degree or above (16.79 ± 4.69)], had debts (16.93 ± 4.46), had no health insurance (17.39 ± 4.38), had no house property (17.82 ± 4.64), had no offspring (17.41 ± 4.65) or fewer siblings [no sibling (16.99 ± 4.94), 1 sibling (16.95 ± 4.47)], or had a history of smoking [smoker (17.11 ± 4.43), ex-smoker (16.84 ± 4.46)] or drinking [drank before 30 days (16.71 ± 4.23), drank in 30 days (17.14 ± 4.51)], showed higher EBS-SF scores than the average score, while the residents who were employed (16.21 ± 4.60), retired (15.05 ± 4.96), married (16.09 ± 4.51), lived in Western China (16.25 ± 4.41) or the urban area (16.59 ± 4.62), or had a monthly household income of 6001–9000 yuan per capita (16.20 ± 4.75), showed lower scores of EBS-SF. The average scores of the other scales were listed as follows: the NGSES scores (28.82 ± 5.39), the Sport Scale scores (7.57 ± 5.46), the PHQ-9 scores (6.22 ± 5.65), the GAD-7 scores (4.49 ± 4.64), the FHS-SF scores (34.99 ± 6.63) and the PSSS scores (60.17 ± 12.90). The following were the mean scores of each personality characteristic on the BFI-10 scale: Extraversion (5.26 ± 1.60), Agreeableness (6.01 ± 1.49), Conscientiousness (5.87 ± 1.61), Neuroticism (4.77 ± 1.50) and Openness (5.46 ± 1.52).

Based on the Ecological Model of Health Behavior, sex, age, personality, and self-efficacy were influencing factors in the personal characteristics of the scores of EBS-SF. Males presented higher EBS-SF scores than females, but there were no significant differences in the EBS-SF scores between males and females. Older residents showed lower scores, and the differences in the EBS-SF scores among age groups had statistical significance. In the second level of the model, individual behaviors played an important part in eating behavior, including drinking, smoking, and emotion-processing. The residents who smoked, drank alcohol, or were depressed or anxious had higher EBS-SF scores in this study. As for the interpersonal networks in the third level, marriage, the number of offspring and siblings, whether living alone, family health, and social support, were all the significant factors affecting eating behavior. The married participants had better eating behavior, while the unmarried respondents showed higher EBS-SF scores. The residents who had offspring showed lower EBS-SF scores, while those with more siblings behaved better in eating. The residents living alone had worse eating behavior. In the fourth level, region, residence, education level, career status, household income, debt, health insurance, and house property posted a great effect on eating behavior. As the results showed, the residents who lived in Western China or lived in the urban area, had moderate household income, had health insurance, had no debt, or had more house properties presented lower scores of EBS-SF. The participants with higher education levels showed higher EBS-SF scores. Moreover, compared with the students and the unoccupied residents, those being retired and employed behaved well in eating.

### 5.2. The Factors Relevant to the EBS-SF Scores

The factors relevant to the EBS-SF scores were listed in Table 3 (see Table 3). The R^2^ in the stepwise regression analysis was 0.254. The VIF of the variables in this stepwise regression was shown in Table A3 in Appendix A with a mean VIF of 1.47 (see Table A3). As the important factors in the first level of the Ecological Model of Health Behavior, females (β = 0.03), higher scores of Extraversion (β = 0.03) or Neuroticism (β = 0.04) on the BFI-10 scale, and higher scores of NGSES (β = 0.09) were contributing factors to the scores of EBS-SF, whereas older age [36–45 years (β = −0.06) or 46–60 years (β = −0.07)], disability (β = −0.03), higher scores of Agreeableness (β = −0.04), Conscientiousness (β = −0.14) or Openness (β = −0.03) on the BFI-10 scale were negative factors to the EBS-SF scores. In the second level of the model, several individual behaviors and emotional-processing also had an impact on eating behavior. The residents who smoked (β = 0.04), drank alcohol in 30 days (β = 0.05), played sports more frequently in one week (β = 0.07), or had higher scores of PHQ-9 (β = 0.30) may have improper eating habits. Marital status and family status were all the significant factors of the interpersonal networks in the third level of the model. The participants who were married (β = −0.04) or had higher FHS-SF scores (β = −0.20) were likely to have better eating behavior, while those who had more than one offspring (β = 0.05) or had more siblings [1 sibling (β = 0.03) or ≥ 2 siblings (β = 0.04)] tended to have unhealthy eating behavior. And the scores of the PSSS had a slightly positive association with the scores of EBS-SF (β = 0.03). At the fourth level of the model, the community level included living region, household income, debt, and career status. The residents who lived in Western China (β = −0.03) had a monthly household income per capita of 6001–9000 yuan (β = −0.04), had no debt (β = −0.02), retired (β = −0.03), presented lower scores of EBS-SF. Compared to other factors, the positive effect of Conscientiousness (β = −0.14) and family health (β = −0.20) was greater on healthy eating behavior, while depression (β = 0.29) had a greater contributing impact on eating behavior related to overweight and obesity.

### 5.3. The Factors Relevant to the EBS-SF Scores Stratified by Sex

The factors associated with the EBS-SF scores of participants stratified by sex were presented in Table A4 in Appendix A (see Table A4). The R^2^ was 0.267 for males and 0.247 for females. The VIF of the variables in the stepwise regression stratified by sex was shown in Table A5 in Appendix A (see Table A5). The mean VIF was 1.47 for males and 1.33 for females.

As the results showed, both male and female residents who were aged 36–60, had a monthly household income of 6001–9000 yuan per capita, had higher scores of Agreeableness and Conscientiousness on the BFI-10 scale or higher scores of FHS-SF would behave better in eating habits, while residents who had two or more than two offspring, drank in 30 days, had higher scores of Neuroticism, NGSES, Sports Scale or PHQ-9 were more likely to present higher scores of EBS-SF. For males, those with a bachelor’s degree or above (β = −0.04) had lower EBS-SF scores, while those who smoked (β = 0.04), had debts (β = 0.03), had two or more than two siblings (β = 0.03), or had higher scores of PSSS (β = 0.06) showed higher EBS-SF scores. Concerning females, the participants who were married (β = −0.04) or retired (β = −0.04), lived in Western China (β = −0.03), or had higher Openness scores (β = −0.04) and lower Extraversion scores (β = −0.03) on the BFI-10 scale presented lower EBS-SF scores. The women with a monthly household income of 3001–6000 yuan per capita (β = 0.03) or with one sibling (β = 0.03) were likely to eat improperly. Among all the factor of males and females, Conscientiousness [male (β = −0.12) or female (β = −0.14)] and family health [male (β = −0.23) or female (β = −0.18)] had a stronger positive influence on eating behavior, whereas depressive emotion [male (β = 0.31) or female (β = 0.28)] had a greater negative impact on healthy eating behavior.

### 5.4. The Factors Relevant to the EBS-SF Scores Stratified by Region

The factors associated with the EBS-SF scores of participants in different regions were presented in Table A6 in Appendix A (see Table A6). The R^2^ was 0.263 in Eastern China, 0.236 in Central China, and 0.306 in Western China. The VIF of the variables in the stepwise regression stratified by region was presented in Table A7 in Appendix A (see Table A7) with the mean VIF of 1.66, 1.34, and 1.61 in Eastern China, Central China, and Western China respectively.

The results showed that residents in retirement or with higher scores of Conscientiousness on the BFI-10 scale or FHS-SF would have lower EBS-SF scores in China, while those who had higher NGSES scores, higher Sports Scale scores, and higher PHQ-9 scores would have higher EBS-SF scores. In Eastern China, participants who were older [36–45 (β = −0.04) or 46–60 (β = −0.06)], had one offspring (β = −0.04), or had higher scores of Agreeableness (β = −0.05) or Openness (β = −0.03) on the BFI-10 scale would behave better in eating. The residents in Eastern China who were female (β = 0.07), smokers (β = 0.03) or ex-smokers (β = 0.04), drank alcohol in a month (β = 0.05), had a monthly household income per capita of 3001–6000 yuan (β = 0.04) or more than 9000 yuan (β = 0.04), had debts (β = 0.04), or had siblings [1 sibling (β = 0.03) or ≥ 2 siblings (β = 0.04)] showed higher EBS-SF scores. Moreover, the participants in Eastern China with higher scores of GAD-7 (β = 0.07) and PSSS (β = 0.04) tended to have worse eating behavior. The participants who lived in Central China at the age of 46–60 (β = −0.07), employed (β = −0.07), or had higher scores of Agreeableness (β = −0.05) on the BFI-10 scale showed lower scores on the EBS-SF. The residents in the central region who were aged 26–35 (β = 0.06), smoked (β = 0.06), lived alone (β = 0.04), had a monthly household income per capita of 3001–6000 (β = 0.04), had more than one offspring (β = 0.06), or had higher scores of PSSS (β = 0.07), were more likely to have improper eating behavior. The respondents in Western China who were at the age of 36–45 (β = −0.07), married (β = −0.19), divorced or widowed (β = −0.05), unoccupied (β = −0.04), lived alone (β = −0.06), had more monthly household income per capita [6001–9000 yuan (β = −0.08) or >9000 yuan (β = −0.06)], or had higher scores of Openness (β = −0.05) on the BFI-10 scale showed better eating behavior. The residents in the western region with offspring [1 offspring (β = 0.11) or ≥2 offspring (β = 0.17)], siblings [1 sibling (β = 0.07) or ≥2 siblings (β = 0.08)], drinking history [drank before 30 days (β = 0.05) or drank in 30 days (β = 0.07)], or higher scores of Extraversion (β = 0.05) or Neuroticism (β = 0.08) on the BFI-10 scale tended to have unhealthy diets. Eating behavior of residents in each region were all greatly influenced by Conscientiousness [Eastern China (β = −0.14), Central China (β = −0.12), Western China (β = −0.15 and family health [Eastern China (β = −0.21), Central China (β = −0.19), Western China (β = −0.21)], and depression [Eastern China (β = 0.25), Central China (β = 0.31), Western China (β = 0.25)]. Moreover, the effect of marriage (β = −0.19) and more offspring (β = 0.17) was greater on eating behavior of residents in Western China.

### 5.5. The Factors Relevant to the EBS-SF Scores Stratified by Places of Residence

The factors associated with the EBS-SF scores of participants in the urban and rural areas were presented in Table A8 in Appendix A (see Table A8). The value of R^2^ in the stepwise regression analysis was 0.261 in the urban area and 0.247 in the rural area. With a mean VIF of 1.48 in the urban area and 1.14 in the rural area, the VIF of the variables in the stepwise regression stratified by residence was posted in Table A9 in Appendix A (see Table A9).

Both urban and rural residents who were male, aged 46–60, had higher Conscientiousness scores on the BFI-10 scale or higher FHS-SF scores would have healthy eating behavior, while those who drank in 30 days, did exercises more often in one week, had higher PHQ-9 scores would have higher EBS-SF scores. The residents living in the urban area who were 36–45 years (β = −0.07), retired (β = −0.04) or married (β = −0.04), lived in the western region (β = −0.03), had a bachelor’s degree or above (β = −0.03), had one house property (β = −0.03) or had higher Agreeableness scores on the BFI-10 scale (β = −0.05) were more likely to do well in eating behavior. However, the urban residents with a monthly household income per capita of 3001–6000 yuan (β =0.06) or >9000 yuan (β = 0.04), with ≥2 offspring (β = 0.05) or siblings [1 sibling (β = 0.03) or ≥2 siblings (β = 0.04)], smoking habit (β = 0.04), or higher scores of Neuroticism on BFI-10 scale (β = 0.05), higher scores of NGSES (β = 0.13) or PSSS (β = 0.04), may have more eating habits related to overweight and obesity. For the rural residents, those who had one offspring (β = −0.06), had ≥2 properties (β = −0.06), or had higher Openness scores on the BFI-10 scale (β = −0.05) tended to have lower scores on the EBS-SF, whereas those who had debts (β = 0.05) used to smoke (β = 0.04), and had higher scores of Extraversion on BFI-10 scale (β = 0.05) were more likely to eat improperly. Among all the influence factors on EBS-SF of both urban and rural participants, the score of Conscientiousness and FHS-SF had a greater negative effect, while depression showed a greater contributing impact.

### 5.6. The Predictive Power of the EBS-SF Score in Differentiating Overweight and Obesity

According to Chinese adult standards, body mass index (BMI) was grouped as underweight (BMI of <18.5), normal weight (BMI of 18.5–<24), overweight (BMI of 24–<28), and obesity (BMI of ≥28) [50]. Among 8623 participants, most people (61.96%) were in the normal weight range, while 1867 respondents (21.65%) were overweight and 353 (4.09%) were obese. Moreover, there were 1060 underweight participants (12.29%). The BMI and the EBS-SF scores of participants are depicted in Table 4 (see Table 4). 

The participants were divided into two groups based on BMI of ≥24 and <24. The residents were also placed into two groups, obesity and non-obesity. The baseline data were comparable. To estimate the predictive capability of the EBS-SF score in differentiating overweight and obesity, ROC curve analysis was conducted on all participants, females and males, respectively (see Table 5 and Figure 3). For the EBS-SF score of all participants, the AUC in predicting overweight or obesity was 0.721, and the AUC in predicting only obesity was 0.731 (See Figure 3I,II). The results of the analysis to estimate the accuracy of the EBS-SF score to discriminate between the female participants with BMI of ≥24 and <24 were statistically significant (AUC = 0.708) (See Figure 3III). The ROC curve was also plotted to assess the predictive accuracy of men’s EBS-SF score in differentiating whether BMI was above the normal range (AUC = 0.740) (See Figure 3V). The AUC of the EBS-SF score for predicting the risk of obesity was 0.736 and 0.729 in females and males, separately (See Figure 3IV,VI). In all, each AUC in this study was greater than 0.70, which is considered a good predictive capacity according to Swets’ criteria. The EBS-SF score can identify the risk of overweight and obesity with a certain degree of accuracy.

## 6. Discussion

In the present study, we analyzed the factors that influenced the eating behavior of Chinese residents aged 18~60 based on the Ecological Model of Health Behavior and validated the predicting power of the EBS-SF score in differentiating overweight and obesity. The average score on the EBS-SF was 16.66. The results showed that several factors influenced the EBS-SF score in each level of the Ecological Model of Health Behavior as follows (See Figure 4):

I: In the first level, sex (H1), age (H2), and personality traits (H3) had complex effects on eating behavior, and females, younger age, and people with more personality of Extraversion and Neuroticism showed worse eating behavior. Self-efficacy was associated with a higher score on EBS-SF (H4). Compared to other factors, the positive effect of Conscientiousness was greater on healthy eating behavior.

II: In the second level, unhealthy individual behaviors such as smoking (H5), drinking (H6), and poor emotional regulation (H8) had negative impacts on eating behavior. Doing exercise more frequently was also related to a higher score of EBS-SF (H7). Depression had a greater contributing impact on eating behavior related to overweight and obesity than other factors.

III: In the third level, marriage (H9) and family health (H10) had positive impacts on eating behavior, while offspring (H10) and siblings (H10) were associated with a higher EBS-SF score. Social support was slightly related to a higher EBS-SF score (H11). Among all the factors in the third level, family health had a stronger influence on eating behavior.

IV: In the fourth level, living in relatively developed areas was a contributing factor to improper eating behavior (H12). Residents in different career statuses presented different eating behavior, among which the retired residents had lower EBS-SF scores (H13). It was worth noting that the middle-income group tended to have a lower score of EBS-SF, though good social-economic status was beneficial to healthy eating behavior (H14). However, the effect of community factors was all lower on eating behavior.

The predictive accuracy of the EBS-SF score in differentiating overweight and obesity was proven with all the AUC greater than 0.70.

### 6.1. Personal Characteristics

In the first level of the Ecological Model of Health Behavior, sex, age, disability, personality, and self-efficacy were significant affecting factors of the eating behavior. Personality was evaluated by the BFI-10. The NGSES was used to measure the self-efficacy of the participants in this study.

#### 6.1.1. Sex

In this study, sex had an impact on eating behavior. Though the average EBS-SF score of women was lower than men, the differences in the average EBS-SF scores between males and females were not significant. Females tended to have a higher EBS-SF score, which was more significant in Eastern China in the stepwise regression stratified by region. Other studies found that emotional eating and uncontrolled eating were positively correlated in females and that women were much more vulnerable to various eating disorders, which was consistent with our results [50,51,52,53]. Therefore, females are supposed to pay more attention to healthy eating behavior.

#### 6.1.2. Age

Age was negatively associated with the EBS-SF score in the present study. Some researchers found that there was a negative association between aging and unhealthy eating behaviors, such as uncontrolled eating and emotional eating [54,55]. The residents aged 26–35 in the central region were more likely to have higher EBS-SF scores in this study. The younger residents may behave worse in eating because of lower inhibitory control, higher reward sensitivity, and higher pleasure-seeking, as it was shown in the research that young adults starting independent life were more vulnerable to developing unhealthy eating habits [51,56]. A study showed that older people would consume more fruits and vegetables than younger adults, and consume more fruits and vegetables instead of high-fat food, which was beneficial to health [57]. Consequently, young adults should lay more emphasis on developing good eating habits, and the aged should maintain healthy eating behavior.

#### 6.1.3. Personality Traits

The results of the Big Five Inventory scale were also influencing factors of the EBS-SF scores. In detail, high Extraversion, low Agreeableness, low Conscientiousness, high Neuroticism, and low Openness were associated with high EBS-SF scores. The association between unhealthy eating behavior and high Neuroticism was in line with previous studies: high Neuroticism was one of the risk factors for emotional eating, and low Neuroticism was related to restrained eating [50,58]. People with higher scores of Openness, Agreeableness, and Conscientiousness were more likely to have a healthier diet in both Chinese and US student samples [59]. The association between the EBS-SF score and Extraversion score in this study was consistent with the results of other studies that suggested that Extraversion included traits that may be linked to obesogenic eating behaviors, such as lower inhibitory control, higher reward sensitivity, and higher pleasure-seeking behavior, leading to extraverted youth being at increased risk of unhealthy diets [60,61]. Moreover, Extraversion was positively associated with food interest [58,62]. Therefore, residents with higher Extraversion, low Agreeableness, low Conscientiousness, high Neuroticism, or low Openness should pay more attention to a healthy diet.

#### 6.1.4. Self-Efficacy

Contrary to expectations, there was a negative relationship between healthy eating behavior and the NGSES score in the study. Interestingly, some studies held different opinions on self-efficacy and eating behavior. For example, Glasofer et al. found that neither BMI, percent fat, carbohydrates consumed, nor snack or dessert intake was related to the general self-efficacy belief of adolescent girls assessed by a 17-item Self-efficacy Scale, but greater general self-efficacy was inversely associated with episodes of lost-of-control eating [63]. The study conducted by Smith et al. showed that there were no consistent effects of parental self-efficacy on children’s ecological momentary assessment outcomes of craving, overeating, and loss of control of eating [64]. Generally, the influence effect and mechanism of general self-efficacy on eating behavior were not fully understood. One potential explanation for the unexpected positive association between the NGSES score and EBS-SF score could be that the participants with higher self-efficacy have more motivational traits or beliefs to keep healthy in several ways though they may tend to have an appropriate amount of indulgences such as taking snacks and desserts.

### 6.2. Individual Behaviors

In the second level of the model, individual behaviors, including lifestyles such as smoking, drinking alcohol, doing sports, and emotional-processing, had an appreciable impact on the eating behavior of residents.

#### 6.2.1. Lifestyle

Lifestyles such as smoking and drinking were associated with high EBS-SF scores. Some studies found that non-smokers had healthier eating behavior than smokers, although nicotine has a suppressing effect on appetite [65,66,67]. The ex-smokers in the eastern region or the rural area were more likely to have unhealthy eating behavior, as abstinence from nicotine would enhance the incentive for food, leading to uncontrolled eating [10]. Alcohol contributed to stimulating appetite and even binge eating, resulting in high EBS-SF scores [68,69]. Higher exercise frequency gave rise to a higher score of EBS-SF to a lesser extent, which was in line with the result, demonstrating the acute effects of exercise on energy intake that exercise could increase hunger and reduce satiety to promote energy intake in a compensatory fashion because of the alterations in hormonal mediators of appetite [70,71]. Therefore, residents should reduce smoking and drinking and be extra mindful of eating a healthy balanced diet after exercise.

#### 6.2.2. Emotional-Processing

Emotional-processing can also influence eating behavior. In the current study, the participants with higher PHQ-9 scores showed higher scores of EBS-SF, indicating the depressed person may have worse eating behavior. The eastern residents with higher GAD-7 scores also presented higher EBS-SF scores. It may be because negative emotions such as depression and anxiety influence the motives for eating and drive the food choice towards more palatable and fewer healthy meals [72,73,74]. Hence, learning emotional regulation, staying in a good mood, asking for help, or consulting a doctor promptly when failing to manage emotions is of great significance to the residents.

### 6.3. Interpersonal Networks

The factors regarding the interpersonal networks in the third level of the model played an important role in eating behavior, including marital status, family status, and social support. The family health was assessed by the FHS-SF. The social support was measured by the PSSS.

#### 6.3.1. Marital Status

As the results showed, the married residents had lower EBS-SF scores than unmarried, divorced, or widowed participants. It was consistent with previous studies that have suggested that marriage was beneficial to healthful eating, while transitions out of marriage influenced dietary behavior to some extent [16,75,76]. In the present study, the EBS-SF score was lower in married women than others with significant differences. This may be because spouses encouraged, monitored, and influenced health behaviors [16].

#### 6.3.2. Family Status

Regarding family factors, though the residents with offspring and more than two siblings had lower average EBS-SF scores, offspring and siblings were positive factors to the EBS-SF score. There was a positive correlation between the number of siblings and the EBS-SF score. It may be because people tended to eat more and even binge in company, and the number of present people was correlated with meal size [77,78]. A higher score of FHS-SF was significantly related to a lower score of EBS-SF in the current study, as social and emotional health, healthy lifestyle, and health resources in the family had an active role in establishing and promoting better-eating behavior.

#### 6.3.3. Social SUPPORT

The association between the PSSS and EBS-SF score was mildly significant in the present study. Another study found that social support and self-efficacy were positively related [79]. The perceived social support and self-efficacy may prevent residents from excessively eating unhealthily while enjoying moderate indulgence to some extent, hence the EBS-SF score was slightly high. However, some studies held the opposite view that perceived social support would be conducive to healthy eating behavior. However, the studies were conducted among the youth aged 14~21 by the Perceived Social Support from the Family Scale and aged 9~15 by the Child Impact Questionnaire, respectively [80,81].

### 6.4. Community Factors

Living area, place of residence, career, education, and the household economy were all the significant community factors influencing the eating behavior of the residents. Social-economic status was reflected by education level, household income, debt, house property, and so on.

#### 6.4.1. Living Region

There exist economic disparities in different regions in China. The eastern region was more developed, while the western region was relatively underdeveloped. The living region had an important effect on the eating behavior of residents, as residents living in relatively developed areas were more vulnerable to eating disorders [82]. Therefore, more attention to eating behavior related to overweight and obesity should be paid by residents in the relatively developed areas.

#### 6.4.2. Career Status

Career status was of great importance to EBS-SF scores in the study. Retirement was a promotive factor in the healthy eating behavior, which was in line with the results showing that retired females had healthier food habits and retired males tended to eat more fruits [83,84]. It could be due to the increased free time of the retired residents that might promote healthier cooking at home [85]. The employed residents also showed lower EBS-SF scores, it may be because the employed participants may have a higher social-economic status for the stable job and the steady income. As some studies showed, obesogenic behavior was more prevalent among people with lower social-economic status, while people with higher social-economic status put more emphasis on healthy eating [86,87].

#### 6.4.3. Education Level

There was a positive association between the education level and the mean EBS-SF score. However, the highest educational level and EBS-SF score presented a related relation in the results of stepwise regression stratified by sex and residence. In detail, the urban residents with an education level of bachelor’s degree or above tended to have better eating behaviors. That could be due to a high education level, contributing to more opportunities for high-paying jobs and steady income, and thus high social-economic status. Male participants with bachelor’s degrees or above showed lower scores of EBS-SF in the current study, which is in line with the studies suggesting that the education level of males was related to healthy eating behavior, as higher education indicated high social-economic status [86,88].

#### 6.4.4. Household Economy

The residents with a monthly household income of 6000–9000 yuan per capita had lower scores of EBS-SF, while those in the low- and high-income groups showed higher EBS-SF scores. The result of the middle-income group eating more healthfully was consistent with other studies suggesting that people with more assets were more likely to be busy at work and have unhealthy eating habits and that a higher prevalence of eating disorders and binge eating occurred among low-income groups [89,90,91,92]. Being in debt, reflecting the lower social-economic status of the resident, was a significant factor for the higher EBS-SF score. The number of house properties, as a reflection of high social-economic status, was a significant factor in the stepwise regression stratified by residence. The urban residents with a house property and the rural residents with more properties had lower scores of EBS-SF in this study, as the number of properties was related to the social-economic status. People with high social-economic status tended to have a healthy diet [86,87].

### 6.5. The Predictive Power of EBS-SF Score in Differentiating Overweight and Obesity

The results of the ROC curve analysis provided preliminary support for the discriminant validity of EBS-SF score in overweight and obesity among Chinese residents aged 18~60 in the present study. This was the first study to validate the EBS-SF in a large sample of the nationwide population in China. The results showed that the EBS-SF, which was designed to assess the eating behavior related to obesity, was suitable for both females and males to not only predict the risk of obesity but also differentiate whether BMI was above the normal range. Thus, we recommended the EBS-SF as an effective supplement to differentiate between normal weight and overweight or obesity.

### 6.6. Strengths and Limitations

In this study, a nationwide sample of the Chinese population aged 18~60 was used to explore the influencing factors on eating behavior for the first time. The results of this study can provide some references and feasible suggestions for conveniently predicting overweight and obesity, reducing the prevalence of overweight and obesity, and promoting healthy behavior. However, this study had several other limitations. As the data of the nationwide sample were cross-sectional, the change in associations could not be studied over time. As a result of the self-reported information and the self-assessed scales in the study, reporting bias may exist. Moreover, we could not know the number of participants who reviewed the online poster or survey but decided not to complete the survey, and thus could not assess non-response bias. Some participants on a weight-loss diet could not be excluded, therefore, careful attention to the possibility of measurement bias and continued evaluation of the EBS-SF are encouraged. Future research could be conducted to explore more factors that may affect the eating behavior of residents based on the current study, such as life stress, social culture, and public policy. Studies on the relations of general self-efficacy, perceived social support, and eating behavior should be also performed in the future. Longitudinal studies should be carried out to assess how the various influencing factors affect the eating behavior of residents. Advocation should be strengthened to improve the healthy eating behavior of residents, especially those who are female, young, extraverted, nervous, depressed, smoking, drinking, living in relatively developed areas, and have debts. Governments should take measures to improve the social-economic status, help residents detect the risk of overweight and obesity early and promote a healthy lifestyle and mental health of the residents to achieve healthy eating in the whole population.

## 7. Conclusions

This study found that several factors influenced the eating behavior of Chinese residents aged 18~60, including personal characteristics, individual behaviors, interpersonal networks, and community factors. Among the influencing factors, Conscientiousness and family health played more promotive roles in healthy eating behavior, while depression had a greater negative impact on eating behavior. As a consequence, residents with risk factors of unhealthy eating behavior, especially depression, should pay more attention to a healthy diet to avoid overweight and obesity, and those with promotive factors are encouraged to keep on pursuing a healthy lifestyle.

The study also found that the score of the short-form of the Eating Behavior Scale had moderate-high predictive accuracy in differentiating overweight and obesity. Thus, this 7-item scale can be used as a reliable and convenient construct to predict the risk of overweight and obesity and promote people to maintain a healthy eating behavior.

## Figures and Tables

**Figure 1 nutrients-14-02585-f001:**
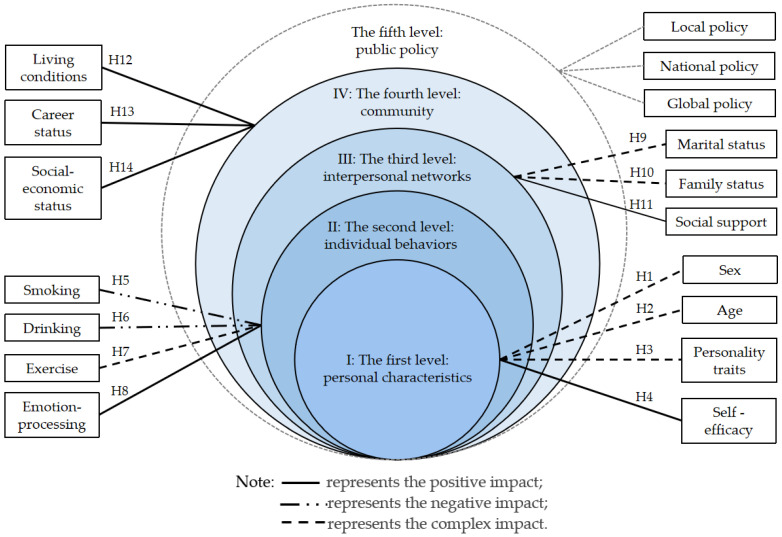
The Ecological Model of Health Behavior and thirteen research hypotheses.

**Figure 2 nutrients-14-02585-f002:**
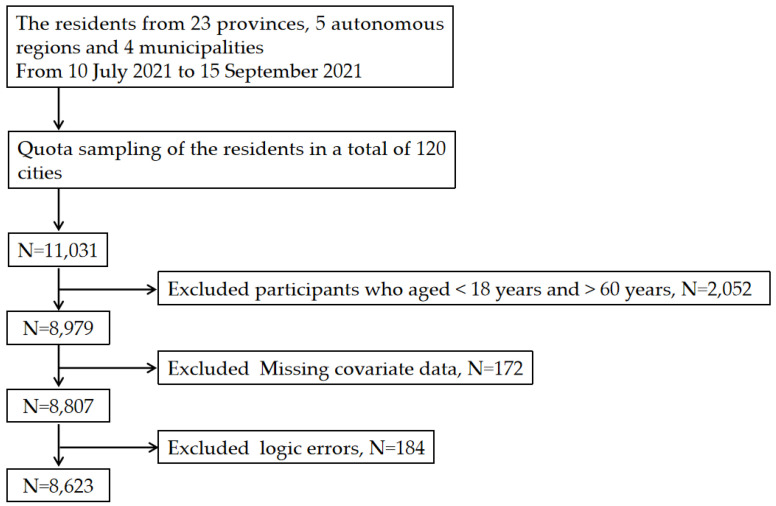
Flowchart of participant enrollment.

**Figure 3 nutrients-14-02585-f003:**
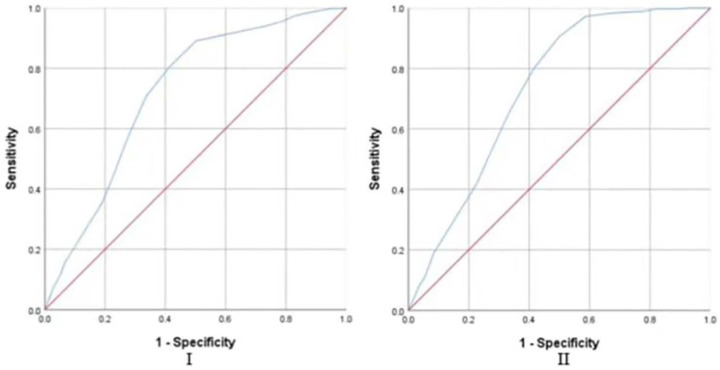
**(I**) ROC curve for the EBS-SF score in predicting the overweight or obesity of all participants; (**II**) ROC curve for the EBS-SF score in predicting the obesity of all participants; (**III**) ROC curve for the EBS-SF score in predicting the overweight or obesity of females; (**IV**) ROC curve for the EBS-SF score in predicting the obesity of females; (**V**) ROC curve for the EBS-SF score in predicting the overweight or obesity of males; (**VI**) ROC curve for the EBS-SF score in predicting the obesity of males.

**Figure 4 nutrients-14-02585-f004:**
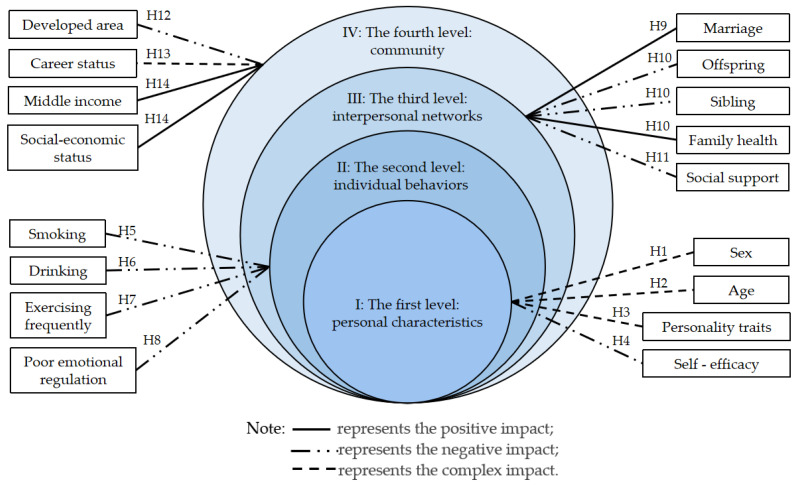
The influencing factors in the Ecological Model of Health Behavior.

**Table 1 nutrients-14-02585-t001:** The statistical description of categorical variables and the EBS-SF scores of study samples.

Categorical Variables	N (%)	the EBS-SF Scores		
		M ± SD	t/F	*p*-Value
Total	8623 (100)	16.66 ± 4.59	—	—
Sex				
Male	3894 (45.2)	16.72 ± 4.60	1.15	0.250
Female	4729 (54.8)	16.61 ± 4.58		
Age group				
18–25	1996 (23.1)	17.71 ± 4.55	84.19	<0.001
26–35	2461 (28.5)	17.02 ± 4.54		
36–45	1998 (23.2)	16.31 ± 4.57		
46–60	2168 (25.1)	15.61 ± 4.42		
Region				
Eastern China *	4492 (51.4)	16.79 ± 4.71	10.13	<0.001
Central China **	2217 (25.7)	16.77 ± 4.48		
Western China ***	1975 (22.9)	16.25 ± 4.41		
Place of residence				
Urban	6454 (74.8)	16.59 ± 4.62	−2.56	0.011
Rural	2169 (25.2)	16.88 ± 4.49		
Highest educational level				
Junior school or below	1449 (16.8)	16.33 ± 4.34	4.20	0.006
Senior school or middle special school	1359 (15.8)	16.53 ± 4.49		
Junior college	1300 (15.1)	16.72 ± 4.55		
Bachelor’s degree or above	4515 (52.4)	16.79 ± 4.69		
Career status				
Student	2134 (24.7)	17.71 ± 4.56	62.90	<0.001
Employed	4548 (52.7)	16.21 ± 4.60		
Retired	220 (2.6)	15.05 ± 4.96		
Unoccupied	1721 (20.0)	16.76 ± 4.28		
Per capita monthly household income, yuan				
≤3000	2328 (27.0)	16.85 ± 4.33	6.77	<0.001
3001–6000	3413 (39.6)	16.68 ± 4.46		
6001–9000	1479 (17.2)	16.20 ± 4.75		
>9000	1403 (16.3)	16.79 ± 5.07		
Whether being in debt				
No	5067 (58.8)	16.47 ± 4.66	−4.59	<0.001
Yes	3556 (41.2)	16.93 ± 4.46		
Whether having health insurance				
No	1752 (20.3)	17.39 ± 4.38	−7.70	<0.001
Yes	6871 (79.7)	16.48 ± 4.62		
Number of house properties				
0	855 (9.9)	17.82 ± 4.64	30.88	<0.001
1	5175 (60.0)	16.56 ± 4.49		
2	2593 (30.1)	16.48 ± 4.69		
Whether living alone				
No	7784 (90.3)	16.55 ± 4.55	−6.78	<0.001
Yes	839 (9.7)	17.68 ± 4.83		
Marital status				
Unmarried	3125 (36.2)	17.60 ± 4.56	108.99	<0.001
Married	5274 (61.2)	16.09 ± 4.51		
Divorced or widowed	224 (2.6)	16.90 ± 4.59		
Number of offspring				
0	3854 (44.7)	17.41 ± 4.65	112.63	<0.001
1	2713 (31.5)	15.73 ± 4.47		
≥2	2056 (23.8)	16.49 ± 4.37		
Number of siblings				
0	2098 (24.3)	16.99 ± 4.94	24.37	<0.001
1	2723 (31.6)	16.95 ± 4.47		
≥2	3802 (44.1)	16.27 ± 4.43		
Whether smoking				
Non-smoker	6907 (80.1)	16.57 ± 4.62	7.40	<0.001
Smoker	1182 (13.7)	17.11 ± 4.43		
Ex-smoker	534 (6.2)	16.84 ± 4.46		
Whether drinking				
No	4870 (56.5)	16.39 ± 4.69	23.34	<0.001
Drank before 30 days	1071 (31.1)	16.71 ± 4.23		
Drank in 30 days	2682 (12.4)	17.14 ± 4.51		
Depression				
No depression	3881 (45.0)	15.22 ± 4.75	364.69	<0.001
Mild depression	3037 (35.2)	16.96 ± 3.88		
Moderate depression	875 (10.1)	18.27 ± 3.58		
Moderate to severe depression	637 (7.4)	19.62 ± 3.14		
Severe depression	193 (2.2)	23.87 ± 4.77		
Anxiety				
No anxiety	4776 (55.4)	15.52 ± 4.65	379.53	<0.001
Mild anxiety	2686 (31.1)	17.35 ± 3.88		
Moderate anxiety	926 (10.7)	19.03 ± 3.33		
Severe anxiety	235 (2.7)	22.66 ± 5.11		

Note: * There are 8 provinces and 3 municipalities directly under the Central Government in Eastern China, including Beijing, Tianjin, Hebei, Liaoning, Shanghai, Jiangsu, Zhejiang, Fujian, Shandong, Guangdong, and Hainan; ** There are 8 provinces of Heilongjiang, Jilin, Shanxi, Anhui, Jiangxi, Henan, Hubei, and Hunan in Central China; *** Western China includes 6 provinces, 5 autonomous regions and 1 municipality of Inner Mongolia, Guangxi, Chongqing, Sichuan, Guizhou, Yunnan, Tibet, Shaanxi, Gansu, Ningxia, Qinghai, and Xinjiang.

**Table 2 nutrients-14-02585-t002:** The statistical description of metric variables.

Metric Variables	M ± SD
EBS-SF scores	16.66 ± 4.59
BFI-10 scores	
Extraversion	5.26 ± 1.60
Agreeableness	6.01 ± 1.49
Conscientiousness	5.87 ± 1.61
Neuroticism	4.77 ± 1.50
Openness	5.46 ± 1.52
NGSES scores	28.82 ± 5.39
Sport Scale scores	7.57 ± 5.46
PHQ-9 scores	6.22 ± 5.65
GAD-7 scores	4.49 ± 4.64
FHS-SF scores	34.99 ± 6.63
PSSS scores	60.17 ± 12.90

**Table 3 nutrients-14-02585-t003:** The stepwise regression analysis of factors associated with EBS-SF scores.

Variables	Coef.	β	*t*	*p*
Sex (Ref: Male)				
Female	0.32	0.03	3.20	0.001
Age group (Ref: 18–25, year)				
36–45	−0.61	−0.06	−4.64	<0.001
46–60	−0.69	−0.07	−5.00	<0.001
Region (Ref: Eastern China)				
Western China	−0.29	−0.03	−2.92	0.003
Career status (Ref: Student)				
Retired	−0.95	−0.03	−3.58	<0.001
Per capita monthly household income (Ref: ≤3000, yuan)				
6000–9000	−0.48	−0.04	−4.27	<0.001
Whether being in debt (Ref: No)				
Yes	0.22	0.02	2.53	0.011
Marital status (Ref: Unmarried)				
Married	−0.34	−0.04	−2.87	0.004
Number of offspring (Ref: 0)				
≥2	0.54	0.05	4.73	<0.001
Number of siblings (Ref: 0)				
1	0.30	0.03	2.61	0.009
≥2	0.35	0.04	2.78	0.005
Whether smoking (Ref: No)				
Smoker	0.50	0.04	3.44	0.001
Whether drinking (Ref: No)				
Drank in 30 days	0.48	0.05	4.64	<0.001
BFI-10 scores				
Extraversion	0.08	0.03	2.98	0.003
Agreeableness	−0.13	−0.04	−3.95	<0.001
Conscientiousness	−0.38	−0.14	−12.14	<0.001
Neuroticism	0.13	0.04	4.05	<0.001
Openness	−0.09	−0.03	−3.20	0.001
NGSES scores	0.09	0.09	8.37	<0.001
Sport Scale scores	0.06	0.07	6.68	<0.001
PHQ-9 scores	0.26	0.29	27.87	<0.001
FHS-SF scores	−0.13	−0.20	−14.69	<0.001
PSSS scores	0.01	0.03	2.44	0.015
Cons.	17.52	0	40.83	<0.001

**Table 4 nutrients-14-02585-t004:** The statistical description of BMI and the EBS-SF scores.

BMI	N (%)		M ± SD	
	Female	Male	BMI	the EBS-SF Scores
Underweight (<18.5)	774 (16.36%)	286 (7.34%)	17.28 ± 1.01	14.93 ± 4.28
Normal weight (18.5~24)	3064 (64.79%)	2279 (58.53%)	21.29 ± 1.50	15.98 ± 4.59
Overweight (24~28)	779 (16.47%)	1088 (27.94%)	25.66 ± 1.10	18.97 ± 3.75
Obesity (≥28)	112 (2.37%)	241 (6.19%)	29.17 ± 0.79	19.97 ± 3.14

**Table 5 nutrients-14-02585-t005:** The results of the ROC curves analysis.

	BMI	N (%)	AUC	*p*	95%CI	
All	Overweight + Obesity	2220 (25.75%) ^a^	0.721	<0.001	0.709	0.732
	Obesity	353 (4.09%) ^a^	0.731	<0.001	0.712	0.75
Female	Overweight + Obesity	891 (18.84%) ^b^	0.708	<0.001	0.691	0.725
	Obesity	112 (2.37%) ^b^	0.736	<0.001	0.704	0.767
Male	Overweight + Obesity	1329 (34.13%) ^c^	0.74	<0.001	0.725	0.756
	Obesity	241 (6.19%) ^c^	0.729	<0.001	0.704	0.753

Note: ^a^ % = N/the number of all participants, ^b^ % = N/the number of females, ^c^ % = N/the number of males.

## Data Availability

The data supporting the conclusions of this article will be made available from the corresponding author upon reasonable request.

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
