# Peer review of "Current Status and Influencing Factors of Eating Behavior in Residents at the Age of 18~60: A Cross-Sectional Study in China"

_nutrients, 2022, doi:10.3390/nu14132585_

Round 1
Reviewer 1 Report
The collected data may lead to an interesting paper if some major issues are addressed.
1. First of all, despite the manuscript is understandable, it has several language issues. Besides requiring an overall review regarding English, some specific points should be addressed:
- The use of "the" is often incorrect, i.e. used when it should't be used and missing where it is required;
- "Eating behaviours" is often used as a synonim of "eating behaviour", which is not correct;
- Throughout the text the authors should replace "gender" with "sex". Gender is a much more complex socia construct, which does not simply correspond to being male or female (it is not a dichotomous feature);
- Several sentences need rephrasing.
2. The introduction does not provide the rationale for all the factors assessed. This is a major issue, since the aims and methods are, therefore, not justified. This is partly addressed in lines 238-255, but simplymoving it to the introduction is not enough.
3. Moreover, the choice of instruments should be explained; for instance, why did the authors choose a general self-efficacy scale instead of a more specific one?
4. In lines 216-217 the stepwise regression model should be further explained, since this may refer to different approaches and criteria.
5. The inclusion of categorical variables in linear regression models, despite sometimes used, is incorrect. As such, the analysis must be redone using a more adequate approach.
6. The full results reported in lines 229-234 should be presented.
7. Some columns presented in table 3 are redundant.
8. As some of the previous comments will lead to possible changes in the results, the discussion must be rewritten accordingly, and will be reviewed in a possible second round.
9. Finally, The first sentence of the conclusions should identify which factors affect eating behaviour, in order to fully answer to the manuscript aims, while the second sentence does not correspond to a conclusion, since it is not a direct answer to the aims (it should be mooved to the discussion).
Author Response
Dear reviewers,
We were pleased to know that our manuscript entitled "Current Status and Influencing Factors of Eating Behavior in Residents at the Age of 18~60: A Cross-Sectional Study in China" was adequately revised.
I really appreciate all your insightful comments and suggestions, which have enabled us to improve our work. Based on the instructions provided in your letter, we uploaded the file of the revised manuscript with all the changes highlighted in red.
Please see the attachment, which is our point-by-point response to the comments and suggestions. The comments are reproduced and our responses are given directly afterward.
We are sure to have satisfactorily improved our manuscript and sincerely hope that it can be accepted for publication. Thanks again for the time and effort that you have put into reviewing our manuscript!
Sincerely,
All authors

Reviewer 2 Report
In this study, Mei et al. seek influencing factors of eating behavior in the Chinese cohort. The topic is good and interesting, and results are tempting. Some data support the well-known knowledge (e.g., smoking and alcohol drinking disrupt eating behavior), and some data are novel and interesting for readers. The manuscript is well written and described. Although data look solid and enough for publication, it would be even stronger if the authors include some clinical status (e.g., diabetes or obesity conditions), and perform ROC curve analysis to show EBS-SF is a powerful tool to distinguish normal and obesity conditions, or other statistical analysis to calculate hazard ratio for each variables associated with obesity supporting that EBS-SF is a reliable score to predict risks of obesity and the government should consider EBS-SF more carefully.
Author Response
Dear reviewer,
We were pleased to know that our manuscript entitled "Current Status and Influencing Factors of Eating Behavior in Residents at the Age of 18~60: A Cross-Sectional Study in China" was adequately revised.
I really appreciate all your insightful comments and suggestions, which have enabled us to improve our work. Based on the instructions provided in your letter, we uploaded the file of the revised manuscript with all the changes highlighted in red.
Please see the attachment, which is our point-by-point response to the comments and suggestions. The comments are reproduced and our responses are given directly afterward.
We are sure to have satisfactorily improved our manuscript and sincerely hope that it can be accepted for publication. Thanks again for the time and effort that you have put into reviewing our manuscript!
Sincerely,
All authors

Round 2
Reviewer 1 Report
I would like to congratulate the authors for the great improvements made to the manuscript. All the issues raised in the previous revision were adequately addressed in this new version.
The explanations and supplementary material now provided justifies the analytic approach, and the discussion focus all the relevant issues. Therefore, and despite what I previously recomended, a new round of revision will not be necessary.
In the current version of the manuscript, the introduction provides an adequate justification to the aims and methods. Bothe the methods and the results are fully described, allowing a better comprehension of the study's contribution. The text was also reviewed regarding English language.
Since all issues were addressed and no new problems were found in this new version, I believe the manuscript may be published in its current form.
Author Response
Point 1:
The authors need to edit the paper for English. For instance, "Based on the data report of the “Seventh National Census in 2021” .... does not make much sense....
Response 1:
Thank you for your useful suggestions. We feel sorry for our poor writing. So we have reread the article and polished some of the paragraphs, specifically modifying and supplementing the vidence and references in the "Materials and Methods" in lines 155-159 on page 4. And the reference data from the official website of National Bureau of Statics in China was authentic and reliable in the present study. We hope the revised manuscript could be acceptable to you. Examples of the modifications are as follows:
“2. Materials and Methods
2.1 Study design
Based on the Chinese population pyramid, quota sampling of the selected residents in 120 cities was conducted with the quota attributes of sex, age, and urban-rural distribution to obtain the samples by sex, age, urban-rural distribution in line with the demographic characteristics [32].
[32]. National Bureau of Statistic. the seventh national population Census. 2021. Availabe online: http://www.stats.gov.cn/ztjc/zdtjgz/zgrkpc/dqcrkpc/ (accessed on 5-11). ”
Point 2:
For instance, "For the past 30 years, the Big Five Inventory (BFI) with 44 short-phrase items has been well proven [31]". First of all there is no such a thing in science as proven, secondly there is only one reference for this statement ... This editor suggest that well proven to be replace by a phrase such as establish and provide at least two more references.....
Response 2:
Thank you very much for your comments. We reread and modified the section of"For the past 30 years, the Big Five Inventory (BFI) ....." you mentioned. In order to increase persuasion about the selection of the 10-item short version of the Big Five Inventory in this study, we supplemented the introduction of the original scale (BFI) and the 10-item short form scale (BFI-10), and provided related references, in lines 206-211 on pages 5-6, as follows:
“For the past 30 years, the Big Five Inventory (BFI) with 44 short-phrase items as the most widely accepted contemporary model of personality was verified in several researches and its five-factor structure has been substantially replicated in several countries [33-35]. The 10-item short version of the Big Five Inventory (BFI-10) was an abbreviated scale of BFI with significant levels of reliability and validity in different versions [36-38].
[33]. John, O.P.; Donahue, E.M.; Kentle, R.L.J.U.o.C.B. The Big-Five Inventory. 1991, 18, 367–385.
[34]. Thalmayer, A.G.; Saucier, G.; Eigenhuis, A. Comparative validity of brief to medium-length Big Five and Big Six Personality Questionnaires. Psychological assessment 2011, 23, 995-1009, doi:10.1037/a0024165.
[35]. Chiorri, C.; Marsh, H.W.; Ubbiali, A.; Donati, D. Testing the Factor Structure and Measurement Invariance Across Gender of the Big Five Inventory Through Exploratory Structural Equation Modeling. Journal of personality assessment 2016, 98, 88-99, doi:10.1080/00223891.2015.1035381.
[36]. Carciofo, R.; Yang, J.; Song, N.; Du, F.; Zhang, K. Psychometric Evaluation of Chinese-Language 44-Item and 10-Item Big Five Personality Inventories, Including Correlations with Chronotype, Mindfulness and Mind Wandering. PLoS One 2016, 11, e0149963, doi:10.1371/journal.pone.0149963.
[37]. Courtois, R.; Petot, J.M.; Plaisant, O.; Allibe, B.; Lignier, B.; Réveillère, C.; Lecocq, G.; John, O. [Validation of the French version of the 10-item Big Five Inventory]. L'Encephale 2020, 46, 455-462, doi:10.1016/j.encep.2020.02.006.
[38]. Rammstedt, B.; John, O.P. Measuring personality in one minute or less: A 10-item short version of the Big Five Inventory in English and German. Journal of Research in Personality 2007, 41, 203-212, doi:https://doi.org/10.1016/j.jrp.2006.02.001. ”